# kHz-precision wavemeter based on reconfigurable microsoliton

Rui Niu[1,2,5], Ming Li[1,2,5], Shuai Wan[1,2], Yu Robert Sun[2,3], Shui-Ming Hu [2,3,4], Chang-Ling Zou [1,2] ✉, Guang-Can Guo[1,2] & Chun-Hua Dong [1,2] ✉

The mode-locked microcomb offers a unique and compact solution for photonics applications, ranging from the optical communications, the optical clock, optical ranging, the precision spectroscopy, novel quantum light source, to photonic artificial intelligence. However, the photonic microstructures are suffering from the perturbations arising from environment thermal noises and also laser-induced nonlinear effects, leading to the frequency instability of the generated comb. Here, a universal mechanism for fully stabilizing the microcomb is proposed and experimentally verified. By incorporating two global tuning approaches and the autonomous thermal locking mechanism, the pump laser frequency and repetition rate of the microcomb can be controlled independently in real-time without interrupting the microcomb generation. The high stability and controllability of the microcomb frequency enables its application in wavelength measurement with a precision of about 1 kHz. The approach for the full control of comb frequency could be applied in various microcomb platforms, and improve their performances in timing, spectroscopy, and sensing.

Benefiting from the ultrahigh quality factor and small mode volume in microresonators, the frequency comb, and the fascinating microsoliton could be generated by the Kerr and Pockels nonlinear optics effects with just mW-level continuous-wave laser[1–6]. For the potential application of microsolitons, especially in optical communication, optical ranging, optical clock, precision spectroscopy, novel quantum light source, spectrometer, and photonic artificial intelligence[7–18], high frequency-stability and tunability of each comb line are strongly desired. As a result of the balance between the dispersion and nonlinear interaction, as well as the drive and dissipation, the frequency difference between adjacent comb lines, which is known as the repetition rate ($f_{\mathrm{rep}}$), is determined merely by the free-spectral range (FSR) of the microresonator. With a stabilized pump laser of frequency $f_{\mathrm{p}}$, we only need to stabilize the microresonator to fix $f_{\mathrm{rep}}$ to realize the full control of the comb frequency.

However, microresonators are vulnerable to the environmental thermal noises and parasitic laser-induced nonlinear effects due to its small mode volume[19–22], and the microsoliton is even more sensitive to the frequency fluctuation of the cavity modes and pump laser for ultra-narrow optical resonance[23]. These deleterious effects can induce local perturbations of the refractive index of the dielectric material and the geometry of microresonator, thus lead to non-uniform shifts to individual optical resonances and also gives rise to the unexpected fluctuation of the FSR. To stabilize the soliton comb, the common cavity tuning approaches that rely on a global change of the dielectric refractive index ($\delta n$) or the cavity geometry ($\delta L$) are utilized based on thermal-optics and electro-optics effects, which lead to a global frequency shift $\delta f/f \propto (\delta n/n + \delta L/L)$[21] of resonances. Such approaches simultaneously tune the frequencies and the FSR of all modes[11,24,25], thus is difficult to realize the full control of the comb. For example, we could not keep the target pump mode wavelength fixed when tuning

[1]CAS Key Laboratory of Quantum Information, University of Science and Technology of China, Hefei 230026, China. [2]CAS Center for Excellence in Quantum Information and Quantum Physics, University of Science and Technology of China, Hefei 230026, China. [3]Department of Chemical Physics, University of Science and Technology of China, Hefei 230026, China. [4]Institute of Advanced Science Facilities, Shenzhen 518107, China. [5]These authors contributed equally: Rui Niu, Ming Li. ✉e-mail: clzou321@ustc.edu.cn; chunhua@ustc.edu.cn

$f_{rep}$, because the whole cavity spectrum is simultaneously shifted by an amount being several orders of magnitude larger than the FSR drift[26]. Then the drive laser would be far-off resonance, and the microsoliton is in a dilemma. Eventually, challenges are imposed to stabilize or even tune the $f_{rep}$, and the practical applications of the microsoliton are hindered.

Here, we propose a universal approach to stabilize and tune the $f_{rep}$ of microsoliton, by using two global-frequency-tuning (GFT) methods simultaneously. In a practical experimental configuration, we introduce a two-temperature model that are independently controlled by a pump laser and an auxiliary laser to realize the self-adaptive stabilization of the microcomb and independent tuning of the $f_{rep}$. We demonstrate the fast, programmable and through frequency controlling of arbitrary comb line, and show a frequency measurement precision around kHz in a proof-of-principle demonstration of wavemeter. Compared to previous work with precision of several MHz[16], the precision of our work achieves a three-order of magnitude improvement. Our work paves the way toward the low-cost and chip-integrated comb spectroscopy and optical frequency standard.

## Results

### Reconfigurable microsoliton

Figure 1a schematically illustrates the basic GFT mechanism of a cavity, as the change of optical round-trip paths can be effectively described by inserting or removing different dielectric materials inside the cavity. In practices, the GFT of microcavity could be induced by the temperature change[26], geometry deformation[27], or electro-optics effect[28], which is almost uniform at the scale of cavity length. As a results of a single GFT, all optical resonances belongs to the same mode family shifts to the same direction [Fig. 1b]. The resonance shifts have slightly

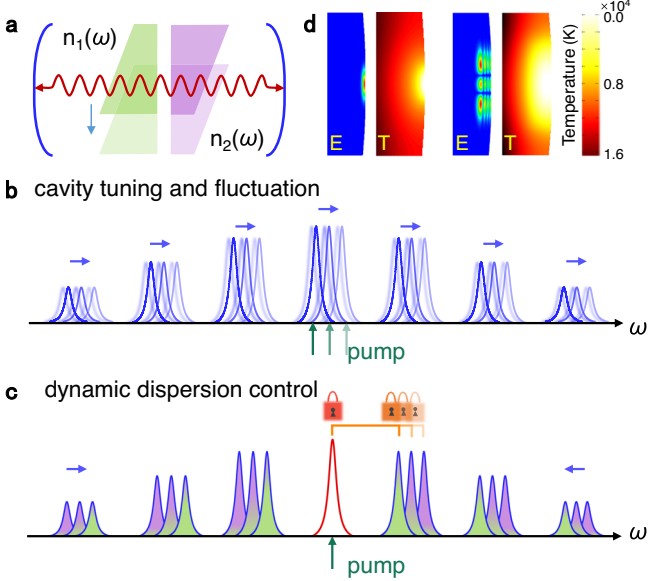

**Fig. 1 | The principle of microsoliton stabilization and tuning by multiple global-frequency-tuning (GFT) methods. a** Schematic of a cavity with two dielectric blocks made of different materials. The frequency of a selected resonance could be fixed by simultaneously controlling the fraction of two blocks, while the free-spectral range of the cavity could be changed due to different effective dispersions for two blocks. **b** The cavity spectral shift under a single GFT. The frequency of the pump laser should follow the resonance shift or fluctuations to sustain a stabilized microsoliton. **c** The cavity spectra under multiple GFT methods. The pump resonance is self-adaptively stabilized while the FSR can be tuned independently. **d** The two-temperature model in a silica microrod cavity. When the cavity is excited by lasers through different spatial modes, the temperature shows distinct spatial distributions and thus induce different tuning effects to optical resonances, corresponding to the multiple GFT methods.

different rates due to the material and geometric dispersions, thus the FSR also changes under a GFT, while the variation of FSR is several orders smaller than that of individual resonance[26]. Therefore, the pump laser frequency $f_p$ and the repetition rate $f_{rep}$ of microsoliton could not be controlled independently by a single GFT method.

However, the real-time controlling of $f_{rep}$ is critical not only for the stabilization of the microsoliton, but also for applications in precision spectroscopy and optical frequency reference. This challenge could be circumvented by introducing distinct GFT methods simultaneously. The tuning could be described as a modification of the effective refractive index of the cavity materials by

$$\delta n_{eff}(f) = \sum_j \frac{\eta_j}{m!} \frac{\partial^m n_j^m}{\partial^m f} (f - f_0)^m,$$  (1)

with $n_j$ and $\eta_j$ denoting the contribution from the $j$-th GFT mechanism and the corresponding weighting factor. $f$ is the frequency and $\partial n_j^m / \partial^m f$ is the dispersion of the corresponding GFT with respect to a reference frequency $f_0$. It should be noted that distinct GFT methods have different dispersion $\partial n_j^m / \partial^m f$, thus it provides the degree-of-freedom $\eta_j$ so that the net dispersion $\partial^m n_{eff} / \partial^m f$ of the cavity could be alternated for $m \geq 1$ under the constraint $\sum_j \eta_j n_j = c$, where $c$ determines the frequency of an individual mode and can be chosen as a constant to match a certain optical reference. For the purpose of solely controlling the $f_{rep}$, we only need two GFT methods to control $\eta_1 \partial n_1 / \partial^m f + \eta_2 \partial n_2 / \partial^m f$ while keep $\eta_1 n_1 + \eta_2 n_2$ fixed, as shown in Fig. 1c.

The experimental demonstration of the independent control of $f_{rep}$ is carried out in a silica microrod cavity. By simultaneously exciting two different spatial modes, two GFT mechanisms are proposed by employing the thermal-optics effect, as shown in Fig. 1d. A drive laser stimulates the microsoliton, and an auxiliary laser is employed to pump another mode with different field distribution. Comparing the calculated electric fields and temperature fields of the fundamental mode for the pump and the high-order mode for the auxiliary laser, the spatial temperature distribution of the auxiliary mode is more uniform than that of the pump mode. Since the thermal-optic refractive index change is proportional to the temperature, the more localized thermal field induced by the pump laser induces a stronger dispersive change of refractive index, i.e., $\partial^m n_1 / \partial f^m \neq \partial^m n_2 / \partial f^m$ and thus induce different GFT with $\eta_j \propto T_j$, with $T_j$ being the temperature increased by the corresponding laser. It is worth noting that, under certain conditions, the frequency of the mode driven by the pump laser could be almost fixed self-adaptively due to the balance between the two-temperatures[29–31]. As a result, with a power and frequency stabilized pump laser, we could tune the FSR of the microcavity by adaptively changing $T_1$ and $T_2$ via varying the power or frequency of the auxiliary laser, while the $\eta_1 n_1 + \eta_2 n_2$ is fixed autonomously, without external feedback control of system (see Supplementary Information).

To certify the flexible mode frequency control via the two-temperature model, we carried out the experiments with the setup shown in Fig. 2a. A pump laser (1551.3 nm) is used to stimulate and sustain the microsoliton, with the assistance of an auxiliary laser driving the microcavity through a different spatial mode from opposite direction, which helps suppressing the strong thermal instability[32]. A typical soliton spectrum (blue lines) is shown in Fig. 2b, with the red lines being the backscattering from the comb generated by the auxiliary laser. For convenience, the comb lines are labeled by integer $\mu$ and the comb frequency is $f_\mu = f_p + \mu f_{rep}$, with $\mu = 0$ corresponding to the pump mode. When the microcavity reaches the soliton state, the frequencies of pump mode family can be effectively tuned by scanning the frequency of auxiliary laser. The numerical results of the mode shifts is plotted in Fig. 2c, indicating the slops of frequency shifts are proportional to the $\mu$, i.e., with the auxiliary laser frequency change while the central mode frequency is fixed. To verify this, we characterized the cavity resonances around soliton comb lines by a weak

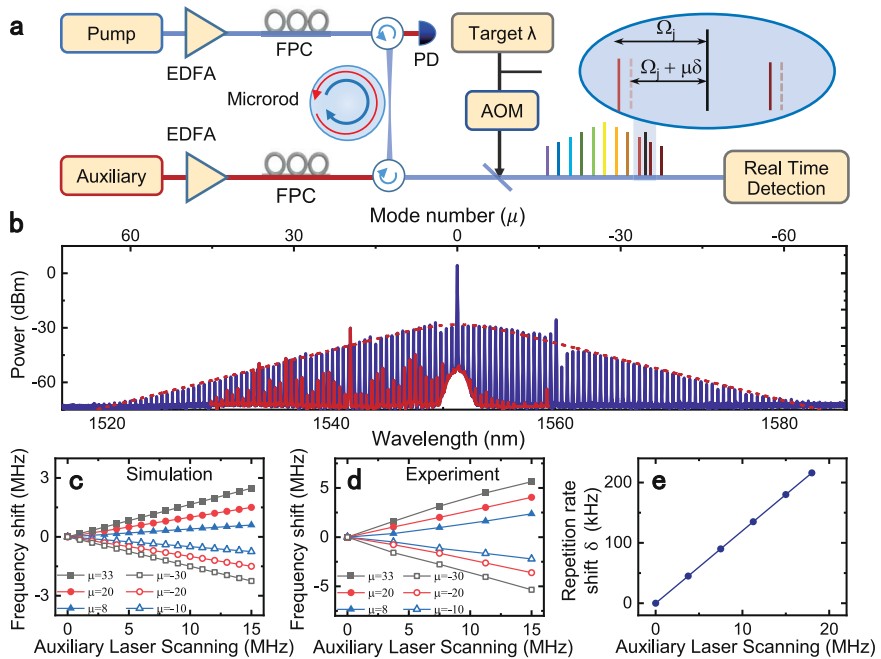

**Fig. 2 | Experimental setup, simulation, and verification measurement.**
**a** Experimental setup. AOM: acousto-optic modulator; FPC: fiber polarization controller; PD: photodetector; EDFA: erbium-doped fiber amplifier. Supplement includes more detail. Inset: The operating principle of the wavemeter. **b** Typical optical spectra of the soliton microcomb. The blue lines represent the comb lines generated by the pump laser and the red lines are due to backscattering of the comb generated by the auxiliary laser. **c, d** The calculated and measured frequency shift of the optical modes for different auxiliary laser frequencies and mode number $\mu$ through the two-temperature model. The measurement is performed on the pump mode family by varying with the auxiliary laser frequency when the microsoliton is kept. **e** Measured repetition rate of the soliton microcomb for different auxiliary laser frequencies.

probe laser (-50 μW)[33], while the single soliton state is sustained and the pump laser is stabilized by a reference cavity (see "Methods"). As shown in Fig. 2d, the measured frequency shifts of the optical modes (see Supplementary Information for more details) are in agreement with the theoretical prediction, and demonstrates that the FSR are indeed tuned by the auxiliary laser. The tuning of FSR also agrees with the linear relationship between the measured shift of $f_{\mathrm{rep}}$ ($\delta$) and the frequency of auxiliary laser [Fig. 2e], which is different from the previous tuning mechanisms based on dispersive wave and Raman self-frequency shift[34–36].

## Operation of wavemeter

The independent tuning of the $f_{\mathrm{rep}}$ by varying the auxiliary laser detuning promises real-time control of the soliton spectrum and enables the locking of $f_{\mathrm{rep}}$ to a RF clock. When the pump is fixed, our device is a reliable frequency reference and enables a wide range of applications, since the frequencies $f_{\mu} = f_{\mathrm{P}} + \mu f_{\mathrm{rep}}$ can be fully determined. As an example, we develop a high-precision wavemeter by the controllable and fully-stabilized microsoliton, which is applied to measure three laser signals, as shown in Fig. 3. The beatnotes $\Omega_j = |\omega_j - f_{\mu}|$ between these lasers $\omega_j$ and the adjacent comb lines $f_{\mu}$ are recorded by the RF spectroscopy. The frequency of the signal is calculated by $\omega_j = f_{\mathrm{P}} + \mu f_{\mathrm{rep}} \pm \Omega_j$, which requires us to determine the value and the sign of $\mu$, and also clear the ambiguity of the sign before $\Omega_j$. To discriminate the sign of $\Omega_j$, we first introduce an acoustic-optics frequency shifter in the measurement setup, by switching the AOM frequency from 80 MHz to 75 MHz and the frequency of the probe laser is changed, thus we could determine the sign by check if the beat note shift is +5 or −5 MHz. Then we sweep the $f_{\mathrm{rep}}$ during the wavemeter operation, and obtain $|\mu| = |\partial\Omega_j/\partial f_{\mathrm{rep}}|$. The sign of $\mu$ is determined by checking the parity of the signs of $\partial\Omega_j/\partial f_{\mathrm{rep}}$ and $\Omega_j$ (see Supplementary Information). The trace of the beatnotes under AOM shifting and $f_{\mathrm{rep}}$ tuning for the wavemeter is shown in Fig. 3a, and the enlarged trace of signals in RF spectra

[Fig. 3a] is presented in Fig. 3c–e, with an AOM frequency switching around the time $t = 6$ s. The corresponding evolution of $f_{\mathrm{rep}}$ in RF spectra is shown [Fig. 3b]. For the example in Fig. 3c–e, we can deterministically derive the orders of the comb lines as $\mu = 18, -9, 17$ for the corresponding probe lasers, respectively. Therefore, the multi-wavelength measurement is simultaneously achieved without ambiguity, which is very challenge for commercial wavemeter. At about 21 s, the repetition rate of the microcomb is tuned suddenly, and the system responses quickly in 1 ms with maintaining the soliton state. We could estimate the locking bandwidth of our method exceeds 1 kHz. In the above operations for realizing the wavemeter, fast tuning and switching of the $f_{\mathrm{rep}}$ are demonstrated, which is unique and could be beneficial for many applications where dual-comb source is required.

## Performance of wavemeter

The performance of our wavemeter is further characterized by measuring a signal with varying frequency. Figure 4a shows a measured pattern USTC in the frequency-time domain by switching the probe laser frequency in real-time. Even the frequency range of the pattern is as small as 1.2 MHz, the pattern can be clearly resolved by our wavemeter, which indicates a high-frequency resolution of the wavemeter. Since the resolution and precision of the wavemeter depends on the frequency stability of the microcomb, the performance can be further improved by locking the $f_{\mathrm{rep}}$ to a microwave reference with feedback to the auxiliary laser. The frequency stability is characterized by the traces of the measured $f_{\mathrm{rep}}$ of the stabilized soliton and the beat note between an ultra-stable laser and the nearest comb line ($f_{\mathrm{beat}}$), as shown in Fig. 4b, c. The $f_{\mathrm{rep(beat)}}$ has an uncertainty of 0.013 (0.49) kHz with a 95% confidence interval. In Fig. 4d, e, the stability of $f_{\mathrm{rep}}$ and $f_{\mathrm{beat}}$ is further tested by the Allan deviations. Comparing the free-running state (hollow orange circle) and the locked state (solid orange circle), our stabilized comb has a significant improved performance, indicating a kHz-level frequency measurement precision (17 kHz at 1 s

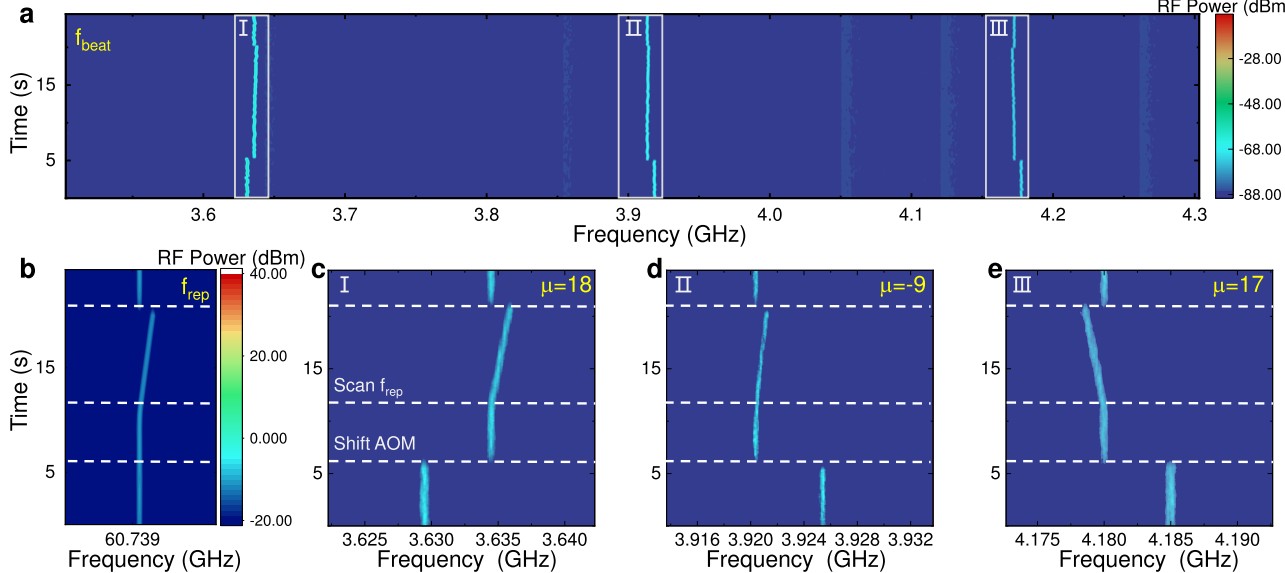

**Fig. 3 | Demonstration of the multi-wavelength measurements. a** The real-time evolution of the beat note between three probe lasers and the adjacent comb lines within 26 s. In this process, the repetition rate is scanned by controlling the auxiliary laser frequency and the probe lasers are shifted by AOM to determine the signs of $\mu$ and $\Omega_j$. **b** The corresponding evolution of the repetition rate in the detection process. **c–e** Detailed evolution of the three beatnotes in (**a**).

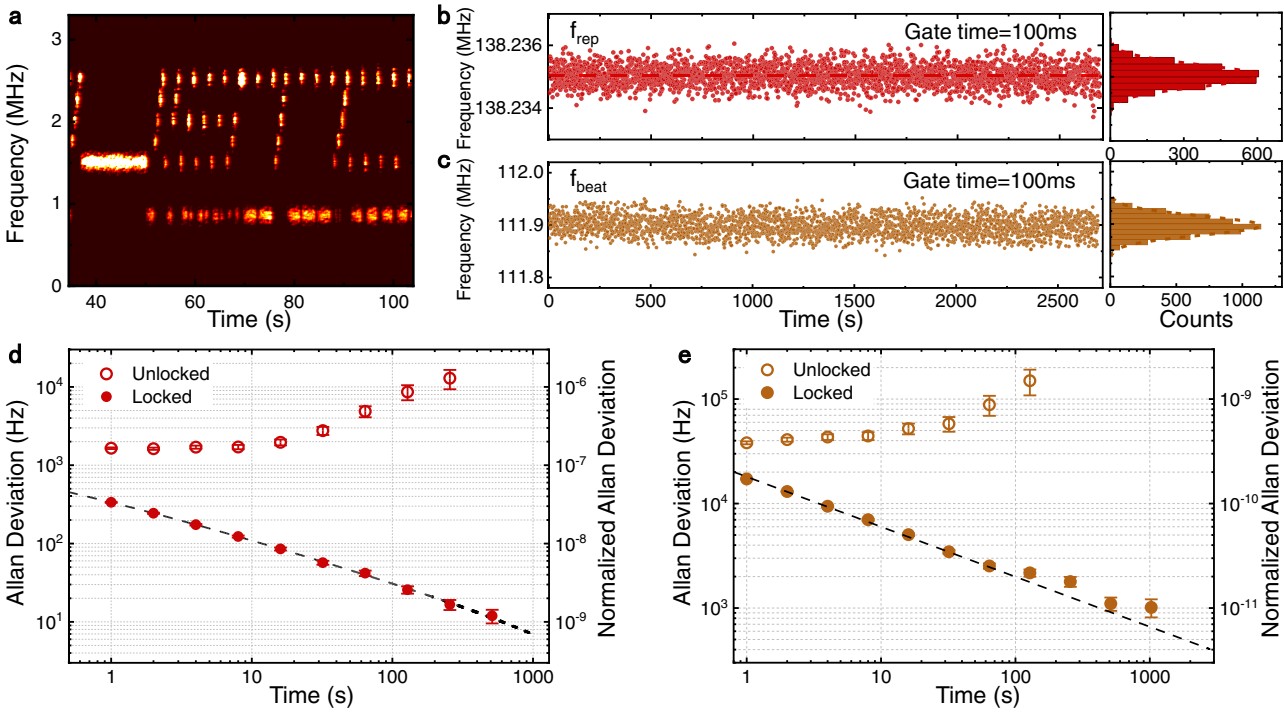

**Fig. 4 | Wavemeter performance. a** The wavemeter measures a probe laser with varying frequency of the pattern USTC. **b**, **c** Time-series measurements of the $f_{rep}$ and $f_{beat}$ when the $f_{rep}$ is locked to a microwave reference with feedback to the auxiliary laser. The gate time of the measurement is 100 ms. **d** Allan deviation of the unlocked (hollow red circles) and locked $f_{rep}$ (solid red circles). At measurement time of 1 s and 512 s, the frequency stability of the locked repetition rate reaches 337 Hz and 12 Hz, respectively. **e** Comparison of Allan deviation of the comb line ($\mu = 17$) for unlocked (hollow orange circles) and locked state (solid orange circles). At measurement time of 1 s and 512 s, the frequency stability of the lock comb line reaches 17 kHz and 1.1 kHz, respectively. Error bars represent a 68% confidence interval.

measurement time). Therefore, the accuracy of $f_p$ in our experiment is inferred as the similar level.

## Discussion

In conclusion, a universal mechanism for the precise and thorough control of microsoliton spectrum is proposed and realized. By introducing multiple GFT methods, individual mode resonance and FSR of a cavity are decoupled and can be tuned independently. In contrary with previous fully stabilized microsolitons[11,24,25], we achieve decouple of the repetition rate and pump laser frequency. Experimentally, the all-optical and self-adaptive control of the microsoliton is realized with a two-temperature model based on the thermo-optic

effect. By switching and stabilizing the microsoliton, a wavemeter with ultrahigh frequency measurement precision at kHz-level and the capability of simultaneously multiple wavelength measurement are demonstrated. The mechanism demonstrated in this work is applicable to all dielectric microcavities with GFT approaches and promises the full control of high-order dispersion of a cavity by introducing more GFT approaches, which might also be useful for comb generation based other nonlinear processes, such as the mode-locked laser and Pockels microcomb. For instance, our scheme could be extended to microring resonators with optomechanical or electro-optics tuning. Therefore, the demonstrated precise microsoliton controlling would facilitate their potential applications in precision measurements, optical clock, spectroscopy as well as communications.

## Methods

### Device fabrication and soliton generation

In this work, the microrod resonator is fabricated from a rotating fused silica rod heated by a focused $CO_2$ laser beam. The diameter of the microrod resonator is around 1.07 mm. The FSR of the microrod is about 60.7 GHz, which agrees well with the repetition rate.

The soliton is generated by using an auxiliary-laser-assisted thermal response control. The pump laser (Toptica CTL 1550) and the auxiliary laser (Toptica CTL 1550) are coupled into the resonator through the tapered fiber from opposite directions with two circulators, and both lasers are amplified by the erbium-doped fiber amplifier (EDFA). The polarization of the pump mode is orthogonal to the polarization of the auxiliary mode. The wavelength of the pump laser is around 1551.3 nm, and the auxiliary is around 1541.72 nm. The input power of the pump laser is around 100 mW, and the power of the auxiliary laser is almost four times higher than that of the pump laser (~380 mW) according to the Q factor of the relevant optical modes. Then, the thermal effect induced by the pump laser is effectively suppressed by the auxiliary laser, which ensures the accessibility of the soliton step in our experiments. By slowly tuning the pump frequency into the pump mode from blue detuning to red detuning, and simultaneously the auxiliary laser offers the knob to realize the self-adaptive cavity tuning presented in this work.

### Stabilization of the soliton

A Pound-Drever-Hall (PDH) frequency stabilization technique was used to lock the frequency of the pump laser relative to the optical mode of a reference cavity, which has a finesse of 250 and a FSR of about 5 GHz. The temperature of the reference cavity is stabilized by a Proportion-Integral-Derivative (PID) servo. By locking the pump laser to the reference cavity, the linewidth of the pump laser is suppressed by ~12 dB (from 1 MHz to 60 kHz).

Limited by the bandwidth of our detector, the comb lines around 1550 nm are filtered and modulated by an electro-optic modulator (EOM) with frequency of 30 GHz to down convert the beat note signal less than 1 GHz. The measured beat note is around 739.05 MHz, corresponding to the repetition rate of 60.739 GHz. Then, the measured beat note signal is phased locked to a reference electronic oscillator (Rohde and Schwarz) through a phase lock loop by feedback to the control current of the auxiliary laser, corresponding to locking the repetition rate to the reference electronic oscillator. Then, the repetition rate can be also tuned by simply changing the reference electronic oscillator. The maximal tuning range of the repetition rate reaches ~200 kHz in our experiment.

### Calibration of the absolute frequency

We calibrate the absolute frequency of the microcomb by referencing a comb line around 1542 nm to a acetylene-stabilized fiber laser (stabi/laser 1542, 194.369489384(5)THz). Based on the stabilized repetition rate and the corresponding $\mu$, we could deduce the absolute

frequency of the stabilized pump laser. Furthermore, we could calculate the absolute frequency of probe lasers.

### Allan deviation measurement

The performance of the stabilized soliton microcomb is characterized by the Allan deviation of the repetition rate and the beat note of the comb line and the acetylene-stabilized fiber laser. The RF frequency is measured in the time domain using a frequency counter (Tektronix FCA 3000) and the Allan deviation is calculated according to

$$\sigma_\nu^2(\tau) = \frac{1}{2(M-1)} \sum_{i=1}^{i=M-1} (\bar{\nu}_{i+1} - \bar{\nu}_i)^2,$$

for different integrating times. Here $\tau$ is the averaging time, $M$ is the sample number of frequency measurements, and $\bar{\nu}_i$ is the average frequency of the signal (measured in unit of Hz) in the time interval between $i\tau$ and $(i+1)\tau$.

## Data availability

All data generated or analyzed during this study are available within the paper and its Supplementary Information. Further source data will be made available on reasonable request.

## Code availability

The code used to solve the equations presented in the Supplementary Information will be made available on request.

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

## Acknowledgements

We thank S.-M. Sun, T.-G. Ma, and J.-M. Cui for discussions and assistance. This work was funded by the National Key Research and Development Program (Grant No. 2020YFB2205801) and the National Natural Science Foundation of China (Grant Nos. 12293052, 92250302, 11934012, 11904316, 11922411, and 11874342), Natural Science Foundation of Anhui Province (Grant No. 2008085QA34) and the Fundamental Research Funds for the Central Universities. M.L. and C.-L.Z. were also supported by the State Key Laboratory of Advanced Optical Communication Systems and Networks. This work was partially carried out at the USTC Center for Micro and Nanoscale Research and Fabrication.

## Author contributions

C.-H.D. and C.-L.Z. conceived the experiments. R.N. and C.-H.D. built the experimental setup, carried out the measurements, and M.L. analyzed the data, with assistance from S.W., Y.R.S., and S.-M.H., C.-L.Z. and M.L. provided theoretical supports. R.N., M.L., C.-L.Z., and C.-H.D. wrote the manuscript, with input from all other authors. C.-H.D., C.-L.Z., and G.-C.G. supervised the project.

## Competing interests

The authors declare no competing interests.
