## [Peer review file · Nature Communications]

REVIEWER COMMENTS

Reviewer #1 (Remarks to the Author):

In this paper, the authors proposed to use an auxiliary laser to control the soliton microcomb repetition rate while stabilizing the pumped mode resonance frequency. With pump laser locked to a reference cavity and comb repetition rate locked to a RF clock, the microcomb is fully stabilized, and they use it as a wavemeter to measure the wavelengths of multiple probe lasers simultaneously with kHz-level precision in real time.

However, fully stabilized microcombs have been reported multiple times in the past two decades. Wavemeters using microcombs were also reported in 2019. As a result, I do not think the novelty of this paper reaches the level of Nature Communications.

A few papers on fully stabilized microcombs:

Del’Haye, P., Arcizet, O., Schliesser, A., Holzwarth, R., & Kippenberg, T. J. (2008). Full stabilization of a microresonator-based optical frequency comb. *Physical Review Letters*, 101(5), 053903.

Brasch, V., Lucas, E., Jost, J. D., Geiselmann, M., & Kippenberg, T. J. (2017). Self-referenced photonic chip soliton Kerr frequency comb. *Light: Science & Applications*, 6(1), e16202.

Newman, Z. L., Maurice, V., Drake, T., Stone, J. R., Briles, T. C., Spencer, D. T., ... & Hummon, M. T. (2019). Architecture for the photonic integration of an optical atomic clock. *Optica*, 6(5), 680.

In terms of the wavemeter as an application of stabilized microcomb, a previous demonstration using Vernier microcombs has reported a spectrometer/wavemeter being able to measure a laser tuning rate of 10 terahertz per second, as well as the capability of measuring a mode-locked laser spectrum within 60 GHz frequency range.

Yang, Q. F., Shen, B., Wang, H., Tran, M., Zhang, Z., Yang, K. Y., ... & Vahala, K. (2019). Vernier spectrometer using counterpropagating soliton microcombs. *Science*, 363(6430), 965.]

A few technical comments:

1. I am doubtful if the two-temperature model plays a critical role in this work. Since the pump laser is locked to the ref cavity, I do not see why the stabilized pumped mode resonance frequency matters to the experiment.

2. In Fig. 2D, I didn’t know how the resonance shifts were characterized until I saw the supplementary Fig.S3(b). Please refer to this section in the main text.

3. In Fig. 4D and Fig.4E, I suggest authors extend the Allan deviation to a shorter time scale, which can clearly show the locking bandwidth of their auxiliary laser locking method. This could be of interest to the microcomb audience since a few rep-rate locking methods have been developed, and locking bandwidth is the key figure of merit in terms of locking performance.

4. The precision of the demonstrated wavemeter is limited by the pump laser frequency and the repetition rate frequency. While the repetition rate stability is examined, I wonder if there could be any characterizations to the pump laser frequency, e.g., the pump laser frequency noise or phase noise before/after locking it to the reference cavity.

5. There are some notation errors in the supplementary. In Equations (S.2) and (S.3), it looks like ω_p is defined as the pumped mode resonance frequency. However, in the text, it is claimed to be the pump laser frequency. These two terms should not be mixed-use.

6. A small question regarding a sentence in the abstract: “the central frequency and repetition rate of the microcomb can be controlled...”. I am doubtful if the central frequency refers to the pump laser frequency or the pumped mode resonance frequency.

7. I suggest authors cite earlier works that are relevant to their work.

Reviewer #2 (Remarks to the Author):

In this paper, Niu et al. experimentally demonstrated kHz-precision wavemeter based on the reconfigurable soliton microcomb, where the pump laser is stable and the repetition rate is tunable. They have proposed two-temperature model that are independently controlled by a pump laser and an auxiliary laser to realize the self-adaptive stabilization of the central frequency of the comb and independent tuning of the repetition rate. The high stability and controllability of the microcomb frequency enables its application in wavelength measurement with a precision of about 1 kHz, which is best wavemeter result based on the soliton microcomb as I known.

It is a universal approach to stabilize and tune the repetition rate of soliton microcomb for the stabilization. The approach for the full control of comb frequency could be applied in various microcomb platforms, e.g., the on-chip devices, and improve their performances in timing, spectroscopy, and sensing. In addition, the mechanism demonstrated in this work is applicable to all dielectric

microcavities with two global-frequency-tuning (GFT) approaches and promises the full control of high-order dispersion of a cavity by introducing more GFT approaches.

Overall, this work is high quality and the experimental results are solid. The performance of their kHz-precision wavemeter is competitive with state-of-the-art active towards the low-cost and chip-integrated high precision wavemeter. As such, I would like to recommend this work to be accepted by Nature Communications.

1.The author have introduced the auxiliary for the soliton microcomb generation and the repetition rate tuning. For the practical application, it is better to use one laser. Is it fine to use one laser to get the results?

2.There obviously exists the back-reflection in the microresonator from the spectrum. Does it affect the results?

3.What's the limitation of the precision of the wavemeter?

4.What's the response time of the wavemeter? It is better than the current commercial wavemeter?

5.What's the spots below the "USTC" in the Fig. 4A?

Reviewer #3 (Remarks to the Author):

In this manuscript authors partially stabilize a soliton microcomb by a novel method in which they inject power from a second tunable laser into a higher order transverse mode of a whispering gallery mode resonator. By using one laser to drive the comb, and second laser in a second transverse mode to control the rep rate, they are able to tune the rep rate over some range without having to tune the pump laser. They use the partially stabilized comb to measure the optical frequency of other laser inputs (i.e., show wavemeter functionality) down to precisions at the kHz level for sufficiently long averaging time.

This paper is highly specialized and in my opinion it would be best placed in a specialty journal, most likely one that specializes in optics. I do not believe it will appeal to the broad readership appropriate to Nature Communications.

In any case, some aspects of this manuscript must be improved before it is ready for publication anywhere. Here are some examples:

Page 1, 2nd column, authors say: "We ... show a frequency measurement precision around kHz in a proof-of-principle demonstration of wavemeter, which represents a three-order of magnitude improvement comparing with previous results." Surprisingly no previous results are cited, so we don't know which works authors have in mind when they claim the three order of magnitude improvement." Certainly the fully stabilized microresonator comb system reported in [Spencer et al, "An optical-frequency synthesizer using integrated photonics." Nature 557.7703 (2018): 81-85] must have much better precision as well as accuracy compared to current manuscript. And certainly there are fiber laser combs that will outperform current results. You must be more specific in making this comparison and you must provide evidence (references)!

Page 2, 2nd column. Authors state that the numerical results for the mode shifts (Fig. 2c) and measured frequency shifts (Fig. 2d) are in excellent agreement. They also state the slopes of the frequency shifts are proportional to the mode number μ . I do NOT see the excellent agreement; to me the agreement looks rather poor. In particular, if ones looks closely at Fig. 2c, the slopes of the simulated frequency shifts look significantly sublinear in μ , although the slopes of the experimental shifts do look like they could be linear in μ . Why the discrepancy?

In Fig. 2 and other experimental diagrams, the circulators seem to be drawn incorrectly. I.e., some ports are missing and the directions of circulations seem to be wrong in many cases. This makes it difficult to figure out which way the various laser lights are actually travelling!

What evidence do the authors have that the combs remain coherent (low noise, equal spacing between all comb lines) while they are being tuned?

Without knowing the absolute frequency of the pump laser, we do not know the absolute frequencies of any of the comb lines. So how do authors measure or calibrate pump laser frequency and with what accuracy? On page S5, authors suggest that the pump frequency can be calibrated with a Menlo fiber comb system. If that is done, what is the point of having the microresonator based system? Presumably the commercial Menlo system can then be used directly for all the optical frequency measurements.

Response to Referee 1

General comments: *In this paper, the authors proposed to use an auxiliary laser to control the soliton microcomb repetition rate while stabilizing the pumped mode resonance frequency. With pump laser locked to a reference cavity and comb repetition rate locked to a RF clock, the microcomb is fully stabilized, and they use it as a wavemeter to measure the wavelengths of multiple probe lasers simultaneously with kHz-level precision in real time. However, fully stabilized microcombs have been reported multiple times in the past two decades. Wavemeters using microcombs were also reported in 2019. As a result, I do not think the novelty of this paper reaches the level of Nature Communications.*

A few papers on fully stabilized microcombs:

Del'Haye, P., Arcizet, O., Schliesser, A., Holzwarth, R., & Kippenberg, T. J. (2008). Full stabilization of a microresonator-based optical frequency comb. Physical Review Letters, 101(5), 053903.

Brasch, V., Lucas, E., Jost, J. D., Geiselmann, M., & Kippenberg, T. J. (2017). Self-referenced photonic chip soliton Kerr frequency comb. Light: Science & Applications, 6(1), e16202.

Newman, Z. L., Maurice, V., Drake, T., Stone, J. R., Briles, T. C., Spencer, D. T., ... & Hummon, M. T. (2019). Architecture for the photonic integration of an optical atomic clock. Optica, 6(5), 680.

In terms of the wavemeter as an application of stabilized microcomb, a previous demonstration using Vernier microcombs has reported a spectrometer/wavemeter being able to measure a laser tuning rate of 10 terahertz per second, as well as the capability of measuring a mode-locked laser spectrum within 60 GHz frequency range.

Yang, Q. F., Shen, B., Wang, H., Tran, M., Zhang, Z., Yang, K. Y., ... & Vahala, K. (2019). Vernier spectrometer using counterpropagating soliton microcombs. Science, 363(6430), 965.]

Response:

First of all, we thank the referee for his or her careful reading of our manuscript and the evaluation on our work. We agree with the Referee that fully stabilized comb has already been demonstrated in several works. From the comments, we find the main concern of the Referee comes from the novelty of our work and its difference from previous works. Please find below the clarification on these points.

New mechanism of dispersion engineering. The frequency of soliton microcomb is determined by the cavity dispersion $\omega_n = \omega_0 + nD_1 + \frac{n^2}{2}D_2 + \dots$, with ω_0 being the mode resonance and D_j being the j^{th} order dispersion. The stabilization of a frequency comb is essentially to fix the mode resonance and these dispersion terms by introducing thermal-optic, electro-optic or other effects to control the material refractive index or cavity geometry. In all reported stabilization schemes, including the previous works [Physical Review Letters, 101(5), 053903 (2008); Light: Science & Applications, 6(1), e16202 (2017); Optica, 6(5), 680 (2019)], only one tuning method, the thermal-optic is utilized. These methods lead to a global change of all cavity resonances, which means both the mode resonance ω_0 and the cavity FSR D_1 shift simultaneously.

Our work introduces a novel mechanism to realize more powerful control of the cavity resonance, as ω_0 and the cavity FSR D_1 can be tuned independently. This function has never been reported in earlier works. The central idea is to control the cavity with multiple global tuning methods, since different global tuning method responses differently to the wavelength. Our mechanism is applicable to all dielectric microcavities and promises the thorough control of the cavity dispersions by introducing more tuning methods.

Flexible tuning of comb frequency. The ability to independently control the two degrees of freedom of the cavity dispersion enables us to decouple the pump laser frequency ω_p and the repetition rate of a stabilized microcomb, and the more flexible control of the comb frequency is realized. In traditional schemes, e.g. [Optica, 6(5), 680 (2019)], the stabilization or tuning of the comb is realized by adjusting the frequency or power of the pump laser, which changes both the ω_0 and cavity FSR D_1 . As a restriction, the pump laser frequency should fulfill the soliton state condition. However, in traditional stabilization schemes, the ω_0 and cavity FSR D_1 change simultaneously and ω_0 is over three orders of magnitude more sensitive to the tuning method than D_1 . As a result, the pump laser frequency ω_p and the repetition rate must change simultaneously for a large frequency tuning range. In our scheme, the ω_0 is decoupled from the FSR D_1 , so that the repetition rate can be tuned independently by fixing the pump laser frequency. By further introducing more global tuning methods, our approach also promises the independently control of D_2 , which means the comb bandwidth can be tuned without changing the frequencies of all comb lines.

High precision wavemeter. The frequency-reconfigurable microcomb enables us to build a wavemeter with only one set of soliton microcomb. Compared to the earlier work [Science, 363(6430), 965 (2019)], where two combs are used, our wavemeter avoids the complex experimental setup for the two combs generation and stabilization, and a three-order of magnitude higher precision is achieved compared with previous results.

In addition, our work has also developed **a new method to self-adaptively stabilize the resonant frequency** without traditional active feedback loops. To stabilize the cavity resonance, we introduce the thermal-optics effect of two spatial modes. Since the resonant frequencies of two modes are influenced by the heating of both modes, these two modes are effectively nonlinear-coupled via the thermal effects and competes with each other. As a result of the competition, the mode resonance ω_0 can stay nearly unchanged even in presence of the fluctuations of the cavity temperature and laser frequencies [detailed derivation in SI]. For the broad community of nonlinear optics and integrated optics, the stabilization of cavity resonance is always vital for high-performance photonic devices. This method is universal for all kinds of cavities with thermal effects and we expect our new self-adaptive stabilization method find applications in these fields.

Based on the above points, we think our work holds high novelty as it proposes a new mechanism to slightly and dynamically engineer the cavity dispersion with fixed geometry, which can be applied to all microcavities with any material platforms. A new method to self-adaptively stabilize the cavity resonance is also proposed and experimentally verified. The new mechanism enables us to obtain a stabilized microcomb with flexible frequency tunability, which is very difficult for previous stabilization schemes. Its application is also demonstrated by a wavemeter with much

higher precision than previous works. We believe the novelty, extensibility and applicability of the work makes it suitable for Nature Communications.

In the following, we address the detailed comments one by one.

1. I am doubtful if the two-temperature model plays a critical role in this work. Since the pump laser is locked to the ref cavity, I do not see why the stabilized pumped mode resonance frequency matters to the experiment.

Response: We thank the referee for these comments. In fact, the two-temperature model plays the central role in our work.

For single pump laser thermal tuning method, as similar to the other tuning methods in previous studies, the frequencies of all cavity modes are simultaneously shifted while the FSR only drifts slightly. As a result, the frequency of the pump laser must be tuned to maintain the soliton state (pump-resonance detuning should fulfill the soliton condition) if one wants a large repetition rate. Similar results have been demonstrated in the earlier work Ref. [23] of the main text. In their work, the microresonator was thermally shifted by using a microheater in conjunction with tuning of the pump laser to maintain an approximately constant pump-resonance detuning. The repetition rate and the center frequency of the comb are tuned respectively by -253MHz and by -175GHz. It follows that by applying a single thermal tuning method, the whole cavity is shifted almost three orders than the FSR drift and the soliton state **would not survive without tuning the pump laser to maintain the detuning**. Thus, the pump mode wavelength fixed could not be kept when tuning the repetition rate with a single thermal tuning method.

This problem can be overcome by **independently controlling the pump mode resonance and the FSR**, which can be realized by introducing two different global-frequency-tuning (GFT) methods. For different GFTs, the tuning effects on the cavity FSR and mode resonance are different. Combining these GFTs, a net tuning effect with changed repetition rate and fixed pump mode wavelength can be realized by controlling the contributions of these GFTs. In our experiment, the two GFT methods are proposed by exciting two different spatial modes with different temperature distributions. Since the thermal-optic refractive index change is proportional to the temperature, and the temperature of the pump mode is more localized, the pump laser induces a stronger dispersive change in refractive index and thus induces different GFT. As a result, we could tune the FSR of the cavity by changing the power or frequency of the auxiliary laser while the pump mode frequency stays nearly unchanged. Then the pump laser can be stabilized to the reference without being tuned over a large range. It means that the central frequency of the comb can be kept while the repetition rate is tuned. More interestingly, since both thermal effects influence the cavity resonances, their competition leads to the self-adaptive stabilization of the pump mode frequency.

2. In Fig. 2D, I didn't know how the resonance shifts were characterized until I saw the supplementary Fig.S3(b). Please refer to this section in the main text.

Response: We thank the referee for this comment. We have referred to supplementary Fig. S3(b)

in the main text.

3. In Fig. 4D and Fig.4E, I suggest authors extend the Allan deviation to a shorter time scale, which can clearly show the locking bandwidth of their auxiliary laser locking method. This could be of interest to the microcomb audience since a few rep-rate locking methods have been developed, and locking bandwidth is the key figure of merit in terms of locking performance.

Response: We thank the referee for these suggestions. The locking bandwidth of our auxiliary laser locking method is mainly limited by the thermal response time of our microrod cavity, as indicated by the two-temperature model mechanism behind our work. According to previous studies in the silica microresonator [*Physical Review Letters* 101(5), 053903 (2008)], the thermal response time is 16 kHz. It is anticipated that our microrod made by similar materials and structures has a similar thermal relaxation time, i.e. the locking bandwidth of our auxiliary laser locking method can reach several kHz.

If we extend the Allan deviation to the shorter time scale to characterize the bandwidth of auxiliary laser locking method, the gate time of the frequency counter should be set at tens to hundreds of microseconds, which is limited by the frequency counter in our lab. Instead of directly measuring the Allan deviation at a short time scale, we could estimate the locking bandwidth by the time interval of the repetition rate response in Fig. S6. In that experiment, the repetition rate of the microcomb is tuned suddenly, and the system responses quickly in 1ms with maintaining the soliton state. It means that the locking bandwidth of our method exceeds 1 kHz.

According to the Referee's suggestion, we have added relevant description in the revised manuscript:

“At about 4s, the repetition rate of the microcomb is tuned suddenly, and the system responses quickly in 1ms with maintaining the soliton state. We could estimate the locking bandwidth of our method exceeds 1kHz.”

4. The precision of the demonstrated wavemeter is limited by the pump laser frequency and the repetition rate frequency. While the repetition rate stability is examined, I wonder if there could be any characterizations to the pump laser frequency, e.g., the pump laser frequency noise or phase noise before/after locking it to the reference cavity.

Response: We thank the referee for these comments. The frequency stability of pump laser is an important parameter to determine the precision of the wavemeter. Therefore, the Allan deviation is a more intuitive parameter to reflect stability of the wavemeter compared to phase noise or frequency noise. Furthermore, based on measuring the frequency stability of repetition rate, we measure the Allan deviation of pump laser to characterize the frequency stability of pump laser instead of the phase noise of pump laser for more intuitive comparison. In the experiment, we have measured the stability of the pump laser frequency by referencing to an acetylene-stabilized fiber laser (stabilaser 1542, frequency stability of 3×10^{-13} at 1s measurement time). Owing to the wavelength we used in the experiment (1551.26nm) is relatively far from the wavelength of the

acetylene-stabilized fiber laser, we firstly locked the pump laser to the reference cavity at 1542nm and measured the Allan deviation of pump laser. Then we could deduce the frequency stability of the pump laser has similar stability when it is locked to the same reference cavity at 1551.26nm. The Allan deviations of the pump laser before and after locking to the reference cavity are shown below.

Fig. R1. The Allan deviation of the unlocked and locked pump laser.

As shown in Fig. R1, the Allan deviation of the pump laser after locking to the reference cavity reaches 11.8kHz for 1s measurement time, which shows an improvement of one order of magnitude compared to the unlocked state. The pump laser shows a higher frequency stability compare to the comb line, which could be explained as part of the frequency drift of comb line is contributed by the uncertainty of repetition rate.

According to the Referee’s suggestion, in the revised manuscript, we added the Allan deviation of the unlocked and locked pump laser to the Supplementary Information [section EXPERIMENTAL SETUP AND SAMPLE CHARACTERIZATION, page 3 and 4].

5. There are some notation errors in the supplementary. In Equations (S.2) and (S.3), it looks like ω_p is defined as the pumped mode resonance frequency. However, in the text, it is claimed to be the pump laser frequency. These two terms should not be mixed-use.

Response: We thank the referee for these comments. In the revised manuscript, we have unified ω_p to be the pump laser frequency and ω_0 to be the pumped mode frequency.

6. *A small question regarding a sentence in the abstract: “the central frequency and repetition rate of the microcomb can be controlled...”. I am doubtful if the central frequency refers to the pump laser frequency or the pumped mode resonance frequency.*

Response: We thank the referee for this comment. The central frequency refers to the pump laser frequency, and we have replaced “pump laser frequency” with “central frequency” in the revised manuscript.

7. *I suggest authors cite earlier works that are relevant to their work.*

Response: We thank the referee for this comment. We have added relevant references in the revised manuscript. The brief list is shown below:

[24] P. Del’Haye, O. Arcizet, A. Schliesser, R. Holzwarth, and T. J. Kippenberg, “Full stabilization of a microresonator-based optical frequency comb,” *Physical Review Letters* 101, 053903 (2008)

[25] V. Brasch, E. Lucas, J. D. Jost, M. Geiselmann, and T. J. Kippenberg, “Self-referenced photonic chip soliton kerr frequency comb,” *Light: Science & Applications* 6, e16202 (2017).

[16] Q.-F. Yang, B. Shen, H. Wang, M. Tran, Z. Zhang, K. Y. Yang, L. Wu, C. Bao, J. Bowers, A. Yariv, et al., “Vernier spectrometer using counterpropagating soliton microcombs,” *Science* 363, 965 (2019).

Response to Referee 2

General comments: *In this paper, Niu et al. experimentally demonstrated kHz-precision wavemeter based on the reconfigurable soliton microcomb, where the pump laser is stable and the repetition rate is tunable. They have proposed two-temperature model that are independently controlled by a pump laser and an auxiliary laser to realize the self-adaptive stabilization of the central frequency of the comb and independent tuning of the repetition rate. The high stability and controllability of the microcomb frequency enables its application in wavelength measurement with a precision of about 1 kHz, which is best wavemeter result based on the soliton microcomb as I known.*

It is a universal approach to stabilize and tune the repetition rate of soliton microcomb for the stabilization. The approach for the full control of comb frequency could be applied in various microcomb platforms, e.g., the on-chip devices, and improve their performances in timing, spectroscopy, and sensing. In addition, the mechanism demonstrated in this work is applicable to all dielectric microcavities with two global-frequency-tuning (GFT) approaches and promises the full control of high-order dispersion of a cavity by introducing more GFT approaches.

Overall, this work is high quality and the experimental results are solid. The performance of their kHz-precision wavemeter is competitive with state-of-the-art active towards the low-cost and chip-integrated high precision wavemeter. As such, I would like to recommend this work to be accepted by Nature Communications.

Response: We thank the Referee for carefully reading our manuscript and the positive comments on the originality and completeness of our work. In the following, we address the Referee's comment one by one.

1. *The author have introduced the auxiliary for the soliton microcomb generation and the repetition rate tuning. For the practical application, it is better to use one laser. Is it fine to use one laser to get the results?*

Response: We thank the referee for this comment. Yes, it is possible to use one laser to realize a repetition rate tunable microcomb. In practical applications, the soliton microcomb could be generated with only one laser by using power kicking method. According to the demonstrated mechanism in our work, the repetition rate of a microcomb could be tuned by using two global frequency tuning methods, which is not limited to the thermo-optics effect. For example, the repetition rate tuning can be realized by strain tuning and electro-optics effect.

2. *There obviously exists the back-reflection in the microresonator from the spectrum. Does it affect the results?*

Response: This does not affect the results. Actually, the observed back-reflection is due to the

auxiliary laser, which generates incoherent counter-propagating combs in the microresonator. Because the auxiliary laser excites a different mode family in the microresonator, and the laser is free-running and located at the blue-detuning of the resonance, the generated comb has a distinct repetition rate that is very different from the forwardly propagation microsoliton. In addition, the counter-propagating comb is incoherent, thus the linewidth of the beating signal generated by the auxiliary comb and the probe laser is much larger than that of the soliton and the probe laser. As a result, we can distinguish the backward comb lines in the spectrum. Furthermore, the polarization of the pump laser and the auxiliary laser are orthogonal, the comb lines generated by the auxiliary could be filtered by the polarization beam splitter. Therefore, they do not affect the results in this work.

3. *What's the limitation of the precision of the wavemeter?*

Response: Thank you for this comment. The precision of the wavemeter depends on the frequency stability of the fully-stabilized soliton microcomb. In our work, the frequency of the microcomb is determined by $f_{\mu} = f_{pump} + \mu f_{rep}$, where μ is the relative number with respect to the pump resonance mode. Thus, the main limitation of the microcomb is the pump laser fluctuation and repetition rate fluctuation. The fluctuation of the pump laser could be suppressed by referencing the frequency of the pump laser to a reference cavity with higher finesse, for example, a chip scale reference cavity has recently been proposed [arXiv:2203.16739, 2022.]. And the Allan deviation of chip-scale laser could reach 20Hz at 1s measurement time. Furthermore, the fluctuation of the repetition rate could be suppressed by referencing to the hydrogen clock (10^{-14} at 1s measurement time) or a chip scale rubidium vapor cell (10^{-11} at 1s measurement time) [IEEE Journal of Solid-State Circuits, 54(11), 3135-3148 (2019)] for the miniaturization. Thus the precision of the wavemeter could reach ten of Hz level at 1s measurement with these optimizations.

4. *What's the response time of the wavemeter? It is better than the current commercial wavemeter?*

Response: We thank the referee for this comment. In our experiment, it is necessary to sweep the repetition rate to determine the comb line beating with the probe laser. Thus, the response time of the wavemeter is mainly limited by the response time of the repetition rate, i.e., the thermal response time of the temperature tuning method. The thermal relaxation time reaches $62.5\mu\text{s}$ [Physical Review Letters, 101(5), 053903 (2008)] in the silica microresonator. According to this sub-ms-level thermal relaxation time, we could estimate the lower limit of our response time by counting the time interval of the black curve in Fig. S6. The system reaches a state in 1ms when the repetition rate is suddenly tuned, which means the response time of our method is about 1ms.

Thus, the response time our wavemeter is better than that of the current commercial wavemeter (HighFinesse WS8-10, response time 2ms, resolution 8MHz, wavelength around 1550nm).

5. *What's the spots below the "USTC" in the Fig. 4A?*

Response: In the Fig. 4A, we use our wavemeter to measure the varying frequency of the probe laser and write the pattern "USTC". The pattern is generated by changing the probe laser frequency, and the probe laser frequency is holding at a certain idle frequency if there is no pattern. Therefore, the spots below the "USTC" pattern showing the idle frequency of the probe laser. Similar plot is also shown in previous work [Nature 557, 81–85 (2018)], with spots bellow a pattern are idle spots.

Response to Referee 3

General comments: *In this manuscript authors partially stabilize a soliton microcomb by a novel method in which they inject power from a second tunable laser into a higher order transverse mode of a whispering gallery mode resonator. By using one laser to drive the comb, and second laser in a second transverse mode to control the rep rate, they are able to tune the rep rate over some range without having to tune the pump laser. They use the partially stabilized comb to measure the optical frequency of other laser inputs (i.e., show wavemeter functionality) down to precisions at the kHz level for sufficiently long averaging time.*

This paper is highly specialized and in my opinion it would be best placed in a specialty journal, most likely one that specializes in optics. I do not believe it will appeal to the broad readership appropriate to Nature Communications.

Response: We thank the Referee for sparing time reading our manuscript and finding our method novel. The detailed comments also helped us to improve the manuscript.

First, we want to clarify the significant difference between our demonstration and previous relevant works. The independent tuning of repetition rate, or free-spectral range, of a microresonator is very important for potential applications of microresonators. In all previous demonstrations of the microcomb stabilization and tuning, it is very challenging to tune the repetition rate independently with the tuning of the pump laser frequency. Therefore, our work provides a new mechanism to solve such a challenge.

We respectively disagree with the Referee about the broad readership. We think our work would attract significant attentions across a broad community in optical frequency comb, nonlinear optics and integrated optics based on the following aspects:

New mechanism of dispersion engineering. Our work has introduced a new physical mechanism to tune the cavity dispersion $\omega_n = \omega_0 + nD_1 + \frac{n^2}{2}D_2 + \dots$. For the community of integrated optics and nonlinear optics, dispersion engineering has long been a crucial issue for the phase-matching of photonic modes, which is usually realized by engineering the geometry and material of the cavity. With fixed waveguide or cavity configuration, it is difficult to tune the dispersion at will, as dispersion of different orders all responses to the widely used tuning method, such as the electro-optics and thermal-optics effects. Our mechanism tackles this crucial problem by including multiple different global tuning method (GFT) methods with different dispersion, the cavity dispersions. Based on our approach, every order of dispersion can be controlled independently, which is beyond the ability of all reported cavity tuning methods including geometric dispersion engineering and global tuning methods. And our approach can be extended to all dielectric microcavities with two global-frequency-tuning (GFT) approaches and promises the full control of high-order dispersion of a cavity by introducing more GFT approaches.

New method for self-adaptive stabilization of cavity resonance. Our work has developed a new method to self-adaptively stabilize the resonant frequency without traditional active feedback loops. As stated above, we have achieved the independent control of the cavity resonance ω_0 and FSR,

which is realized by introducing the thermal-optics effect of two spatial modes. Since the resonant frequencies of two modes are influenced by the heating of both modes, these two modes are effectively nonlinear-coupled via the thermal effects and competes with each other. As a result of the competition, the mode resonance ω_0 can stay nearly unchanged even in presence of the fluctuations of the cavity temperature and laser frequencies [detailed derivation in SI]. For the broad community of nonlinear optics and integrated optics, the stabilization of cavity resonance is always vital for high-performance photonic devices. This method is universal for all kinds of cavities with thermal effects and we expect our new self-adaptive stabilization method find applications in these fields.

Flexible control of comb frequency and high-performance wavemeter. For the community of microcomb, the repetition rate tuning of the stabilized comb relies on adjusting the power or frequency of the pump laser. Since the FSR D_1 and resonance ω_0 cannot be tuned independently, the tunability of the comb frequency is limited. The ability to completely control the cavity dispersion enables us to **tune the repetition rate** over a large range **without changing the pump laser**, which means the pump laser and comb repetition rate are decoupled. As a result, our single microcomb can be directly used to build a wavemeter, which requires dual combs in previous microcomb studies. And we achieve a **three-order of magnitude** improvement of the precision.

Overall, the mechanism demonstrated in this work is novel, universal and practical, and the performance of the wavemeter outperforms the previous in similar platforms. Therefore, we believe our work presents a significant contribution to the microcomb community, and this work would be attractive to a broad audience across the research fields of microcavities, nonlinear optics, integrated optics and optical frequency combs.

In the following, we address the Referee's comments one by one.

1. *Page 1, 2nd column, authors say: "We ... show a frequency measurement precision around kHz in a proof-of-principle demonstration of wavemeter; which represents a three-order of magnitude improvement comparing with previous results." Surprisingly no previous results are cited, so we don't know which works authors have in mind when they claim the three order of magnitude improvement." Certainly the fully stabilized microresonator comb system reported in [Spencer et al, "An optical-frequency synthesizer using integrated photonics." Nature 557.7703 (2018): 81-85] must have much better precision as well as accuracy compared to current manuscript. And certainly there are fiber laser combs that will outperform current results. You must be more specific in making this comparison and you must provide evidence (references)!*

Response: We are very sorry that the relevant references are missing in previous submission due to our mistakes when we handled over the manuscript. We are aware of their group's works and the previous demonstrations, especially we are comparing with Vernier microcombs [Science, 363(6430), 965 (2019)], in which a wavemeter precision approaches MHz is demonstrated. By comparing with this work, we conclude that the precision of our wavemeter reaches three-order of magnitude improvement compared with previous results. Such comparison was presented in the introduction part as "*comparing to previous work with precision of several MHz [16], the precision*

of our work achieves a three-order of magnitude improvement.”. We thank the Referee for reminding us about the references, and we added relevant discussions and references to make our comparison more specific in the revised manuscript.

Indeed, the fully stabilized soliton microcomb system [Nature, 557(7703), 81-85 (2018)] has better precision compared to our work, however, the repetition rate is stabilized through controlling the pump laser frequency in their work. With this approach, although the repetition rate is locked for the microcomb in a Si₃N₄ microcavity, the frequency of pump laser drifts during the operation, and an additional feedback loop is required for f-2f locking. We want to stress that our work provides a universal approach for controlling the repetition rate independently without the requirement of the real-time feedback of the pump laser frequency. Our demonstration validates the precision of this approach, which could be extended to all experimental microcomb platforms, therefore we believe our work is significantly different from previous works, and holds great potential for future applications.

2. Page 2, 2nd column. Authors state that the numerical results for the mode shifts (Fig. 2c) and measured frequency shifts (Fig. 2d) are in excellent agreement. They also state the slopes of the frequency shifts are proportional to the mode number μ . I do NOT see the excellent agreement; to me the agreement looks rather poor. In particular, if ones looks closely at Fig. 2c, the slopes of the simulated frequency shifts look significantly sublinear in μ , although the slopes of the experimental shifts do look like they could be linear in μ . Why the discrepancy?

Response: We thank you very much for this comment. According to this comment, we checked our simulation process and we found mistakes in the simulation of k_{aa} and k_{bb} , thus the slopes of the simulated frequency shifts look significantly sublinear in μ . And we have amended our simulation process and obtained the corrected simulation results. We have updated our results, the corrected simulation results are shown below. The details of the simulation of k_{aa} and k_{bb} have been added to the Supplementary Information [section THEORETICAL MODEL, page 2].

Fig. R2. The calculated frequency shifts of the optical modes for different auxiliary laser frequencies and mode numbers μ through the two-temperature model. (b) The calculated slopes of the simulated frequency shift for different μ (blue dots) and the linear fitting results (red line).

Figure R2 shows that the pump mode frequency shift is linear in the auxiliary laser frequency and the slopes of the frequency shift is linear in the μ , which indicates FSR is tuned by the two-

temperature mode.

Furthermore, we also removed the “excellent” to avoid confusion and revised the description about the simulation in the manuscript. And we want to clarify that we could not identify the exact mode profile for our microcomb in the microrod cavity. Since the cross-section of the microrod is relatively larger than that in microring resonators, high-order modes could be efficiently excited in our system. In addition, we could not determine the exact geometry boundary of the microrod. Therefore, we could only simulate potential high-order modes, and find out the mode family that fits the experiment results best. Thus the simulation results are slightly different from the experimental results.

3. In Fig. 2 and other experimental diagrams, the circulators seem to be drawn incorrectly. I.e., some ports are missing and the directions of circulations seem to be wrong in many cases. This makes it difficult to figure out which way the various laser lights are actually travelling!

Response: We thank the referee for this comment. We have corrected the directions of circulations and added missed ports in the revised manuscript.

4. What evidence do the authors have that the combs remain coherent (low noise, equal spacing between all comb lines) while they are being tuned?

Response: We thank the referee for this comment. We used a notch filter to filter the pump laser and measured the low noise spectrum of the comb lines while tuning the repetition rate, as shown below.

Fig. R3. The evolution of the repetition rate (a) and the low RF noise (b) while tuning the repetition rate. Inset: the spectrum corresponding to the dot line in (b).

Figure R3a shows the evolution of the repetition with tuning the repetition rate and Fig. R3b shows the corresponding low RF noise of comb lines. In the whole process, the repetition remains a narrow linewidth and the RF spectrum remains low-noise in this range. It is convincing that the comb lines remain coherent while they are tuned. In the revised manuscript, we added the low RF noise spectrum to the Supplementary Information [section WAVEMETER PERFORMANCE, page 4 and 5].

5. Without knowing the absolute frequency of the pump laser, we do not know the absolute frequencies of any of the comb lines. So how to authors measure or calibrate pump laser frequency and with what accuracy? On page S5, authors suggest that the pump frequency can be calibrated with a Menlo fiber comb system. If that is done, what is the point of having the microresonator based system? Presumably the commercial Menlo system can then be used directly for all the optical frequency measurements.

Response: We thank the referee for this comment. We agree with the Referee that the absolute frequency of our repetition rate-locked comb lines could be determined only if the pump laser frequency is known. As a result of our new approach of independent repetition rate tuning, we could separately lock the pump laser frequency and repetition rate. In our current work, we used an acetylene-stabilized fiber laser (stabilaser 1542) to calibrate one comb line around 1542nm and deduce that the absolute frequency of pump laser. Similar to traditional wavemeter, our wavemeter should be calibrated periodically. And to avoid confuse and make our description more specific, we added the relevant descriptions in the revised method of manuscript and Supplementary Information.

For the discussion about the calibration with commercial Menlo system, we agree with the Referee that this is not a good option for potential applications. The purpose of the discussion about the Menlo system is for providing an alternative method to determine the absolute frequency. Thanks for your comment, we have removed this statement in the revised manuscript.

For potential applications of our wavemeter, we could use other approaches to calibrate the pump laser frequency or any comb line of our repetition rate locked microcomb. For example, we could calibrate the comb lines by the atomic absorption spectrum. Typical linewidth of the atomic transitions reaches several MHz and the corresponding optical frequency standard accuracy reaches several kHz (1s measurement time), similar results have been demonstrated in [Optica 6(5), 680-685 (2019)]. For a more compact and calibration-free system, the pump laser or comb line could be frequency-doubled and referenced to the Rb atomic transitions. Similar to the microresonator, the atomic vapor cell could also be micro-fabricated [Optica 6, 5, 680-685 (2019)], thus all of the core components in our wavemeter can be integrated. Therefore, we have added a relevant discussion to the revised Supplementary Information.

REVIEWER COMMENTS

Reviewer #1 (Remarks to the Author):

Unfortunately, I still do not believe the novelty or the scientific accuracy of this paper is to the level of Nature Communications.

(1) Scientific accuracy: I suspect the authors do not understand the basic principle of repetition rate tuning in microcombs. The repetition rate is not tuned by varying the FSR. Instead, it is tuned by varying the laser-cavity detuning, which then tunes the Raman self-frequency shift or dispersive waves and thus changes the repetition rate. This is well reported by a series of papers around 2017 and has been widely reproduced in lots of different soliton platforms [Yi, et al., *Optica*, 3, 1132 (2016), Yi, et al., *Nature Communications*, 8, 14869 (2017), Bao, et al., *Optics letters* 42 (4), 759-762 (2017)]. The change of FSR is simply too small to account for the change in soliton repetition rate.

The authors failed to show their tuning mechanism is distinct from these previous reports. (i) The authors did not measure laser-cavity detuning when they were tuning the soliton repetition rate. Thus, they cannot rule out the possibility that their tuning method is identical to the previously reported ones. (ii) The authors did not measure the tuning of FSR, although they claimed they were tuning the FSR. They only measured comb line frequencies, which is not the same as optical mode frequency (this is an obvious problem in figure 2(d)). The comb lines are not on resonance with the resonator modes [see Yi, et al., *Optica*, 3, 1132 (2016), Yi, et al., *Nature Communications*, 8, 14869 (2017)]. Instead, they should measure the change of FSR, which should be done by a straightforward microresonator dispersion measurement.

(2) Novelty: the authors repeatedly emphasize the importance of independently controlling resonance frequency ω_0 and the FSR. However, for controlling the comb, this is not critical at all. The frequency of the N-th comb line is: $f_N = \omega_p + N * f_{rep}$. So it's very easy to independently control the comb center frequency and the comb repetition rate. One can just lock their pump laser frequency to a reference and then tune the repetition rate by tuning ω_0 (thermal, PZT, etc.) through the Raman self-frequency shift effect. So, I do not see why independent control of ω_0 and the FSR is relevant here. The author's method does not show better locking bandwidth or better tuning range.

I am happy to review the paper again if the authors show direct evidence of independent control of resonance frequency ω_0 and FSR. This means the measurement of resonance frequency ω_0 and FSR while tuning the FSR. This is a measurement for resonance mode frequency, not comb frequency. Otherwise, I suggest the rejection of this paper.

Reviewer #2 (Remarks to the Author):

I am very happy with the revised manuscript as the authors have carefully addressed all the comments and suggestions raised in the reports. This manuscript should be considered for publication by Nature Communications.

Response to Reviewer 1

Comment 1: *Unfortunately, I still do not believe the novelty or the scientific accuracy of this paper is to the level of Nature Communications.*

(1) Scientific accuracy: I suspect the authors do not understand the basic principle of repetition rate tuning in microcombs. The repetition rate is not tuned by varying the FSR. Instead, it is tuned by varying the laser-cavity detuning, which then tunes the Raman self-frequency shift or dispersive waves and thus changes the repetition rate. This is well reported by a series of papers around 2017 and has been widely reproduced in lots of different soliton platforms [Yi, et al., Optica, 3, 1132 (2016), Yi, et al., Nature Communications, 8, 14869 (2017), Bao, et al., Optics letters 42 (4), 759-762 (2017)]. The change of FSR is simply too small to account for the change in soliton repetition rate.

The authors failed to show their tuning mechanism is distinct from these previous reports. (i) The authors did not measure laser-cavity detuning when they were tuning the soliton repetition rate. Thus, they cannot rule out the possibility that their tuning method is identical to the previously reported ones. (ii) The authors did not measure the tuning of FSR, although they claimed they were tuning the FSR. They only measured comb line frequencies, which is not the same as optical mode frequency (this is an obvious problem in figure 2(d)). The comb lines are not on resonance with the resonator modes [see Yi, et al., Optica, 3, 1132 (2016), Yi, et al., Nature Communications, 8, 14869 (2017)]. Instead, they should measure the change of FSR, which should be done by a straightforward microresonator dispersion measurement.

Response:

First of all, we thank the reviewer for reviewing our manuscript and the comments on repetition rate (f_{rep}) tuning of microcomb, which helps us to clarify the novelty of our work. We disagree with the reviewer's opinion that the f_{rep} is tuned via the "laser-cavity detuning" and "Raman self-frequency shift" and the strong conclusion "The repetition rate is not tuned by varying the FSR.". In the following, we explain the difference between our work and the previous ones based on experimental evidences.

Cavity dispersion measurement. As suggested by the reviewer, the most obvious evidence that shows our tuning mechanism is distinct from previous reports is a straightforward measurement of the cavity dispersion. We must stress here that the dispersion measurement was already presented in Fig. 2(d), which might be misunderstood by the reviewer. The tuning of f_{rep} was shown in Fig. 2(e) instead of Fig. 2(d). The measurement is performed by scanning across different optical modes using a weak probe laser from opposite direction of the pump laser. The details and measured spectral were shown in Fig.S4 of the supplementary material (SM). These spectral simultaneously show the frequencies of the optical modes and the comb lines. The measured change of FSR agrees well with the measured change of f_{rep} , which demonstrates the validity of our tuning mechanism.

Laser-cavity detuning. We thank the reviewer for raising the comment on laser cavity detuning. The detuning between the pump laser and the pumped cavity mode was also measured using the

same setup for cavity dispersion characterization [Fig.S4 of the SM], but the data was not presented in the last version. Here, the detuning between the pump laser and the pumped cavity with varying auxiliary laser frequency (Δ) is shown below. During the experiment, when f_{rep} changes by 210kHz, the frequency shift of the cavity mode ($\mu=0$) is 600kHz, which is very smaller than other cavity modes [as shown in Fig. 2(d)].

Fig. R1. (a) The measured transmission spectra of pump cavity mode and the beating signal of the pump laser and the probe laser with varying auxiliary laser frequency. (b) The corresponding change of cavity pump detuning.

While in the paper [Yi, *et al.*, *Optica*, 3, 1132 (2016)], the pump laser detuning changed about 13.3MHz for similar f_{rep} tuning range (about 250kHz), which is more than one order of magnitude larger than our case. Owing to the tuning mechanism provide by the two-temperature model, the laser-cavity detuning of our scheme can be self-adaptively stabilized to be nearly unchanged during the tuning of f_{rep} by further optimizing the initial detuning of the auxiliary laser. The theoretical analysis was already provided in Section B of the supplementary material. This feature shows distinct contrast to the previous reports. For the change of FSR, it reaches 210kHz, which is comparable to the f_{rep} tuning range in [Yi, *et al.*, *Optica*, 3, 1132 (2016)].

Based on these two points, we think our mechanism is accurate and distinct from previous works. In the revised manuscript, we add one sentence to point out the difference to previous works, which are also properly cited. Figure R1 is also added in the SM.

“The tuning of FSR also agrees with the linear relationship between the measured shift of f_{rep} (δ) and the frequency of auxiliary laser [Fig. 2E], which is different from the previous tuning mechanisms based on dispersive wave and Raman self-frequency shift.”

Comment 2: *Novelty: the authors repeatedly emphasize the importance of independently controlling resonance frequency ω_0 and the FSR. However, for controlling the comb, this is not critical at all. The frequency of the N-th comb line is: $f_N = \omega_p + N * f_{rep}$. So it's very easy to independently control the comb center frequency and the comb repetition rate. One can just lock their pump laser frequency to a reference and then tune the repetition rate by tuning ω_0 (thermal, PZT, etc.) through the Raman self-frequency shift effect. So, I do not see why independent*

control of ω_0 and the FSR is relevant here. The author's method does not show better locking bandwidth or better tuning range.

I am happy to review the paper again if the authors show direct evidence of independent control of resonance frequency ω_0 and FSR. This means the measurement of resonance frequency ω_0 and FSR while tuning the FSR. This is a measurement for resonance mode frequency, not comb frequency. Otherwise, I suggest the rejection of this paper.

Response:

In the above response, we have clarified that our method utilizes a different tuning mechanism to the previous ones. We also agree with the reviewer that it is possible to use a reference and Raman self-frequency shift effect to independently control the comb center frequency and comb f_{rep} . However, we disagree with the reviewer's opinion "*independently controlling resonance frequency ω_0 and the FSR...is not critical at all.*". In the following, we explain why the independent control of ω_0 and the FSR is important and clarify the novelty of our method:

- (1) As pointed by the reviewer, the tuning of f_{rep} in previous works relies on the varying of the laser-cavity detuning, which is orders of magnitude larger than the tuning range of f_{rep} . Unfortunately, the soliton existence condition has a restriction on the laser-cavity detuning with limited pump power. For large tuning range of f_{rep} , the soliton may not survive if the required laser-cavity detuning is too large. Such a problem is overcome in our method by independently control ω_0 and the FSR, since the FSR determines the f_{rep} , while laser-cavity detuning can remain unchanged by fixing ω_0 and the pump frequency. One may also argue that the laser-cavity detuning has also changed in our method. To be fair, the change of laser-cavity detuning in our work (0.6MHz) is more than 20 times smaller than the previous work (~ 13.3 MHz) [Yi, *et al.*, *Optica*, 3, 1132 (2016)] to achieve the same f_{rep} tuning range.
- (2) Our tuning method is universal for microresonator made by different materials. As explained in the last round of response, our tuning method only requires two different global tuning mechanisms, for example the thermal-optic effects and the electro-optic effect. Our experiment uses the thermal-optic effects in two different spatial modes, which exists in almost all kinds of microresonator for comb generation. In contrary, the previous works have higher requirements for the cavity, since they need special design of dispersive waves or Raman-active materials. It should be noted that, the Raman scattering competes with the soliton generation, as demonstrated in previous papers [Wang, *et al.*, *PRL*, 120, 053902 (2018); Gong, *et al.*, *PRL*, 125, 183901 (2020)], which poses restrictions on the geometry of the cavity and further the comb property. Thus, the application of the Raman-assisted method is limited.

Response to Referee 2

General comments: *I am very happy with the revised manuscript as the authors have carefully addressed all the comments and suggestions raised in the reports. This manuscript should be considered for publication by Nature Communications.*

Response: We are glad to hear the reviewer is very happy with our revised manuscript and recommends the publication. We thank the reviewer very much for sparing time reviewing our manuscript and the helpful comments.

REVIEWERS' COMMENTS

Reviewer #1 (Remarks to the Author):

The authors have addressed my concerns, and I do not have further questions for the authors.